# Emotion Analysis and Dialogue Breakdown Detection in Dialogue of Chat Systems Based on Deep Neural Networks

Kazuyuki Matsumoto [1,*], Manabu Sasayama [2], Minoru Yoshida [1], Kenji Kita [1] and Fuji Ren [1]

[1] Graduate School of Sciences and Technology for Innovation, Tokushima University, Tokushima 770-8506, Japan; mino@is.tokushima-u.ac.jp (M.Y.); kita@is.tokushima-u.ac.jp (K.K.); ren@is.tokushima-u.ac.jp (F.R.)

[2] Department of Information Engineering, National Institute of Technology, Kagawa College, 551 Kohda, Takuma-cho, Mitoyo-shi 769-1192, Japan; sasayama@di.kagawa-nct.ac.jp

[*] Correspondence: matumoto@is.tokushima-u.ac.jp

**Abstract:** In dialogues between robots or computers and humans, dialogue breakdown analysis is an important tool for achieving better chat dialogues. Conventional dialogue breakdown detection methods focus on semantic variance. Although these methods can detect dialogue breakdowns based on semantic gaps, they cannot always detect emotional breakdowns in dialogues. In chat dialogue systems, emotions are sometimes included in the utterances of the system when responding to the speaker. In this study, we detect emotions from utterances, analyze emotional changes, and use them as the dialogue breakdown feature. The proposed method estimates emotions by utterance unit and generates features by calculating the similarity of the emotions of the utterance and the emotions that have appeared in prior utterances. We employ deep neural networks using sentence distributed representation vectors as the feature. In an evaluation of experimental results, the proposed method achieved a higher dialogue breakdown detection rate when compared to the method using a sentence distributed representation vectors.

**Keywords:** natural language processing; dialogue breakdown; human-computer dialogue system; sentiment analysis; emotion recognition

## 1. Introduction

A number of recently-created chat dialogue systems based on artificial intelligence techniques have the ability to generate flexible response sentences [1–4]. However, breakdowns often occur during dialogues using these systems. There are, in fact, various types of dialogue breakdowns. Arend et al. [5] investigated breakdowns in human–robot interactions in a case study analyzing dialogue breakdowns in conversations between humans and the robot that was used in the study. Our study targets the detection of dialogue breakdowns by analyzing utterances between dialogue systems such as a chatbot and users of the system.

Many of the methods proposed for dialogue breakdown detection [6–10] are based on word meaning or dialogue acts, changes of topic, etc. Few attempt to use emotional changes or an emotional feature to detect dialogue breakdown, although, in reality, humans are often made to feel uncomfortable, not by a discrepancy in topics, but rather by a lack of consideration of emotional changes or emotional change patterns. The following dialogue sentences can be used to illustrate. 'S' indicates an utterance made by a system; 'U' indicates an utterance made by a user of the system.

- S: "Where did you buy those clothes?"
- U: "I bought them at the ** department store. They are my favorite."
- S: "They look cheap."

The dialogue flow shown here could occur in real life. However, the expression of such a negative opinion of the user's "favorite clothes" would very likely be taken as an insult and hurt the user's feelings, causing the dialogue to break down. Although there is no dialogue breakdown in either context or topic here, this sort of dialogue flow should be avoided in order to smoothly move the conversation forward. Moreover, even though the dialogue system in our example may be generally useful, the kind of problem illustrated here may well discourage the user from using the system further in order to avoid being upset by its response.

To conduct a smooth and intimate chat dialogue with a user, it is important for a system to have the ability to agree and sympathize with the user's utterances according to the user's emotional changes, in addition to being able to focus on the topics of the dialogue. In this study, we propose a method for detecting dialogue breakdown by comparing emotional tendencies that can be estimated from the utterances of the system and the user. To accomplish our objective, we constructed an emotion level estimator capable of estimating the strength of each emotion based on an utterance distributed representation vector. A dialogue breakdown label is estimated by calculating the similarities among the emotion estimation results obtained by the system from the target utterance, the emotion estimation results from the user's most recent utterance, and the emotion estimation results by the system from the utterance prior to the user's most recent utterance; using this emotion estimator, we then use combinations of the similarities as the feature. We also propose another method using feature quantities that are obtained from an emotional level vector based on emotion estimation results from both system and user utterances prior to the target utterances. In an evaluation experiment, we estimate dialogue breakdown labels using the proposed methods and compare the results with those of other methods, including a method based on a similarity vector for a sentence distributed representation and a current state-of-the-art method.

As our experimental dataset, we used the development data of a chat dialogue corpus that was collected in a Project Next NLP dialogue task [11]. This corpus consists of data from recorded dialogues between a chat dialogue system and a user, with annotated dialogue breakdown labels and comments by several annotators [12]. The annotations indicated three types of dialogue breakdowns: "O", "T", and "X"; however, detailed categories for each breakdown were not annotated. Higashinaka et al. [13,14] categorized the various kinds of dialogue breakdowns. The meanings of the dialogue breakdown labels are shown below.

- O: Not a breakdown: it is easy to continue the conversation.
- T: Possible breakdown: it is difficult to continue the conversation smoothly.
- X: Breakdown: it is difficult to continue the conversation.

In this study, we focus on breakdowns caused by a lack understanding of emotion, which might correspond to a "lack of sociability" or a "lack of common sense" under the broad breakdown category of "environment" proposed by Higashinaka et al. However, sense-based factors such as emotion are not defined in Higashinaka's categorizations.

In Section 2, we introduce related research on dialogue breakdown detection, emotion recognition from dialogue text, and utterance intention recognition in dialogue. In Section 3, we describe the proposed emotion estimation method, the extraction of similar pattern features, and an estimation method for dialogue breakdown labels. In Section 4, we present the evaluation experiments, and the results of the proposed emotion estimation method and dialogue breakdown detection method. Section 5 provides a discussion based on the experimental results, whereas Section 6 concludes the paper.

## 2. Related Works

In this section, we introduce previous research relating to dialogue tearing detection methods, emotion recognition for dialogue texts, the recognition of speech intentions in dialogue, and examples of research related to interaction. Moreover, the relevance and differences between these studies and the present study are discussed.

### 2.1. Method of Dialogue Breakdown Detection

The baseline system that was distributed in the dialogue breakdown detection challenge [15] uses conditional random fields (CRFs) and considers the labels of the system utterances immediately preceding each system utterance. It takes into account the label of the previous system utterance for each system utterance. The user utterance is assigned the label PREV-O/PREV-T/PREV-X as the dialogue breakdown label of the previous system utterance, which indicates whether the previous system utterance has not broken down, is likely to break down, or is broken down. In the Dialogue Breakdown Detection Challenge, most participating teams used rule-based methods, support vector machines (SVMs), and deep learning methods. In particular, the majority of the participating teams used deep neural network methods.

In the SVM-based method, the word vectors of the system utterance and previous user utterances are used as features. Another method is to learn the word vectors using a recurrent neural network (RNN) or LSTM encoders based on the dialogue behavior of the target sentence and previous utterance as features. Rule-based methods are mainly based on keywords that are extracted by morphological analyzers and exhibit low versatility.

Hori et al. [16] provided an overview of the experimental setup and evaluation results of the 6th Dialogue System Technology Challenge (DSTC6), which aims to develop an end-to-end dialogue system. In it, it is mentioned that Track3's dialogue breakdown detection technology performed as well as humans in both English and Japanese languages. However, since the dialogue breakdown detection techniques that show the best performance on a particular dataset often end up being specialized for that dataset, there is still room for improvement in terms of generalizability.

Tsunomori et al. [17] selected measures for dialogue breakdown detection in Dialogue Breakdown Detection Challenge 3. In addition to Accuracy, which is the percentage of correct labels, they considered several other measures, including Accuracy (NB, PB + B) and Accuracy (NB + PB, B). As a result of their selection, they concluded that the RSNOD (NB, PB, B) scale is the best, based on the evaluation of the stability of the rankings and the system discrimination. This measure calculates the Root Symmetric Normalized Order-Aware Divergence (RSNOD) for each system utterance from the distribution of the output of the dialogue breakdown detector and the distribution of correct answers.

Using the Nao communication robot, Maitreyee et al. [18] investigated how young and older adults perceive the actions, roles, and goals of dialogue participants in response to dialogue breakdown.

Kontogiorgos et al. [19] analyzed the multimodal behavioral responses of humans to robot dialogue breakdowns in various tasks. In their study, they found that consistent nonverbal behavior is exhibited in response to certain dialogue breakdowns by robots.

Takayama et al. [20] proposed a method for dialogue breakdown detection by training a dialogue breakdown detector based on a set of annotators grouped using clustering and ensembling multiple detectors. Their method performs global and local dialogue breakdown detection using LSTM, and significantly improves the performance of the Dialogue Breakdown Detection Challenge 3 (DBDC3) over the conventional baseline method using conditional probability fields.

### 2.2. Emotion Recognition from Dialogue Text

Görer [21] proposed an automatic emotion classification method that combines dialogue act modeling and natural language processing approaches, taking into account the temporal flow of the conversation. Our method differs in that we do not model dialogue acts, but use the results of sentence-by-sentence sentiment estimation for dialogue breakdown detection.

Ren et al. [22] proposed a method for correctly recognizing emotions in dialogue by taking into account long-term experience, rich knowledge, and complex patterns of context and emotional states. In their method, they proposed a new concept, the KES model, to enhance semantic information using external knowledge. This model considers both

the external knowledge and the semantic role elements of the conversation, and realizes emotion recognition considering both internal and external states.

Chen et al. [23] used a dictionary of emotional expressions for automatic classification on an unlabeled Chinese emotion corpus and used it for supervised learning. They experimented with a text corpus automatically generated using automatic speech recognition (ASR) from speech signals in the Chinese audio–visual database (CHEAVD), and achieved a performance improvement of more than 10% over the baseline. They use oversampling to balance the training of class imbalanced data. In our work, our goal is different in that we aim to improve performance by considering emotion in dialogue breakdown detection rather than the accuracy of emotion estimation results.

Gao et al. [24] constructed a new dataset called Emotional RelAtionship of inTeractiOn (ERATO), which contains multimodal information of video, audio, and text, based on dramas and movies in order to build a model for recognizing emotions from videos. They proposed a model consisting of synchronous modal–temporal attention (SMTA) units for multimodal fusion to perform the task of pairwise emotion relationship recognition (PERR). Since their method proposes a method for recognizing emotions from multimodal information, it is different from our research on textual dialogues.

*2.3. Intention or Interaction Recognition in Dialogue*

Song et al. [25] proposed a multi-label classification method for recognizing multiple intentions as a method for recognizing speech intentions for concise and non-standardized linguistic expressions. In their study, they reported that BERT features achieved the best results in an evaluation experiment using the Chinese Multi-Intentional Dialogue Dataset (CMID-Transportation). If the intentions of the utterances can be correctly estimated, dialogue breakdowns can be detected by detecting utterances that do not match the intentions. However, in the case of Japanese conversation, especially in colloquial speech, the omission of subjects and objects frequently occurs, and polysemous words and expressions with similar meanings, but that need to be used differently depending on the situation, are frequently used. Thus, recognizing multiple speech intentions is a very difficult task. In this study, we aim to detect dialogue breakdown without recognizing speech intentions.

Wei et al. [26] focused on Interaction Style, which classifies the way we interact with others, and proposed a method to classify four types of Interaction Style based on the obtained Interaction Style profile information using Support Vector Machines. Since the method of interaction varies depending on the characteristics of the user and the dialogue system, we believe that building an interaction style recognition model that is specific to each type of interaction would be useful for detecting dialogue breakdowns. However, in our data, the dialogue system is fixed and the user is assumed to be an unspecified person. Therefore, a large amount of dialogue data must be collected and a corpus annotated with interaction styles is required for modeling. In this study, we assume that the discomfort caused by the difference in interaction style causes a change in the user's emotion, and use the result of emotion estimation as the feature value.

Other methods include Saha et al.'s [27] proposed method for classifying dialogue acts using a multi-task network that takes into account emotions.

## 3. Proposed Method

The proposed method calculates the similarity between the emotion level vectors estimated for the response sentence of the system and the user utterance, and builds a dialogue breakdown label estimation model using the combinations as similarity pattern features. We consider the situation where the system predicts the dialogue breakdown when it generates the response sentence. Therefore, we use the system's current (target for breakdown detection) utterance and the system's and the user's utterances instead of using the user's utterance after the system's current utterance. The flow of the proposed method is illustrated in Figure 1.

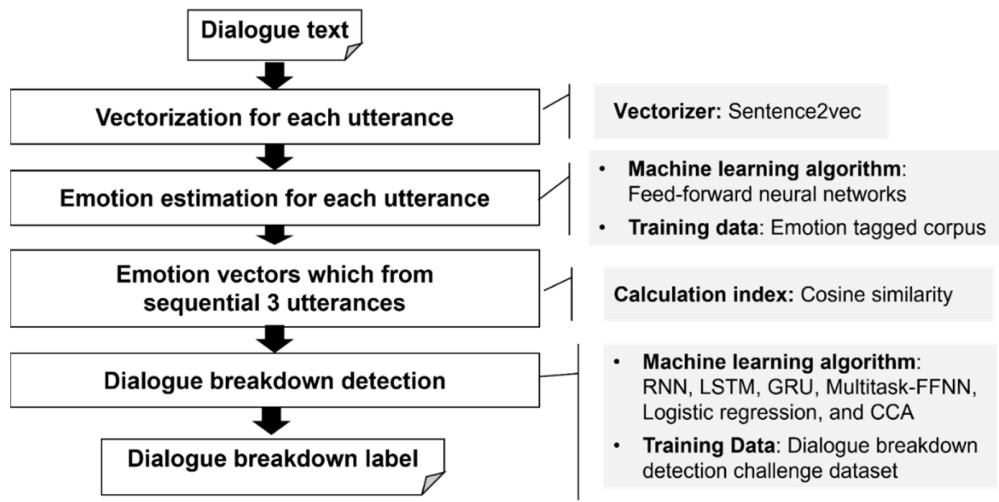

**Figure 1.** Flow of proposed method.

### 3.1. Target Corpus

In this study, we use the dataset that was used in the dialogue breakdown detection challenge, and to which the dialogue breakdown label was assigned, as the target of analysis. This data is a corpus of text recordings of one-on-one dialogues between a human and a dialogue system (chatbot), which were labeled by several third parties as either broken, not broken, or in between. The columns (O,X,T) are the labels of dialogue breakdown. A total of 30 annotators assigned labels for O (not a breakdown), T (possible breakdown), and X (breakdown). For example, if (O,X,T) = (11,9,10), there are 10 annotators who are labeled as broken down. User (U) is a human speaker and has not been given a broken dialogue label, so all values are set to −1. The details of the data are described in detail in the dataset section in Section 4. Table 1 shows the example of the dataset of the dialogue with breakdown labels.

**Table 1.** Example of the dialogue with dialogue breakdown labels.

| S/U | Utterance | (O, X, T) |
|-----|-----------|-----------|
| S | Good evening. Be careful with heat stroke. | (30, 0, 0) |
| U | Thank you. | (−1, −1, −1) |
| S | Okay. | (11, 9, 10) |
| U | How about that? | (−1, −1, −1) |
| S | You must be careful with heat stroke. | (5, 11, 14) |
| U | Yes. And I got you a heat stroke at first sight. | (−1, −1, −1) |
| S | Heat stroke is okay. | (8, 8, 14) |
| U | Which way are you talking about? | (−1, −1, −1) |
| S | Prevention is important. | (6, 12, 12) |

### 3.2. Emotion Estimation

The pre-processing of emotion estimation requires the construction of a vector generating model that converts sentences, phrases, and words into vectors. To perform this task, we used sentence2vec [28], an implementation of paragraph2vec proposed by Le and Mikolov [29,30]. We adopted the paragraph2vec algorithm, which can express the similarity between paragraphs or sentences by converting texts into a dense vector trained using neural networks. In a study conducted by Le and Mikolov [31], the paragraph2vec algorithm achieved state-of-art results in the sentiment analysis and text classification tasks. Figure 2 shows the framework for training the paragraph vector.

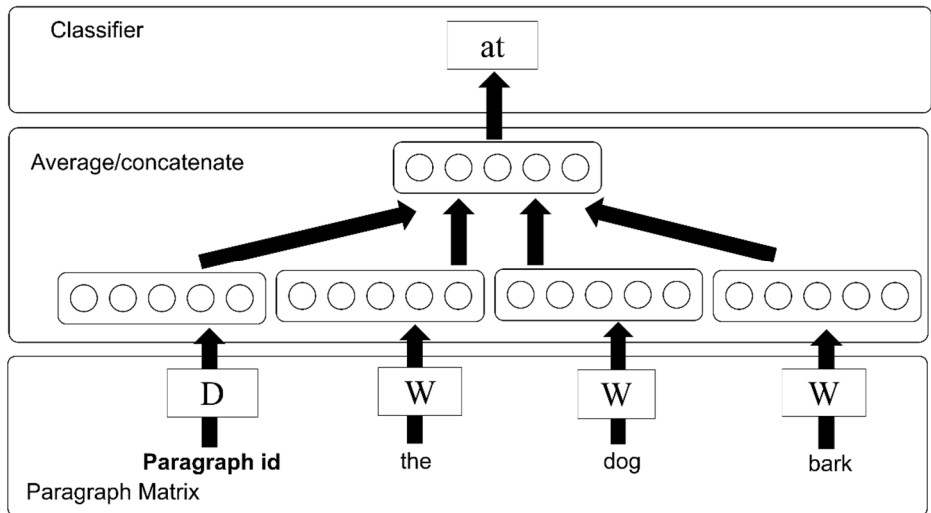

**Figure 2.** Framework for training the paragraph vector.

### 3.2.1. Sentence Embedding

We converted semantically similar/related sentences into similar vectors. The sentence vectors were trained using a vector dimension size of 500 and a context window size of 10. Generally, the larger the vector dimension size, the more detailed the information that can be expressed. Because the context window size represents the number of peripheral context words, the larger the context window size becomes, and the wider the relations that can be considered.

The randomly collected 3,403,658 tweets were tokenized by the Japanese morphological analyzer MeCab [32] and used to create the vector generation model.

### 3.2.2. Neural Networks

To construct the emotion estimation model, we used an utterance corpus consisting of 42,273 spoken sentences and an emotional expression dictionary containing 19,529 emotional expressions (words or phrases) with annotations indicating emotion type. Based on a systematic chart proposed by Fischer [33], six kinds of emotion labels were annotated to the utterance sentences included in the corpus. Several utterances and emotional expressions were annotated with more than one label. These labels were weighted according to their levels of strength. Table 2 shows the various emotion labels and the number of annotations to our emotional corpus. The example sentences included in the corpus are shown in Table 3. The number of examples, the number of words, and the number of vocabularies are shown in Table 4.

**Table 2.** Labels and number of annotations.

| Label | Example of Emotions | Num. of Annotations |
|-------|---------------------|---------------------|
| A-1 | Joy, relief, sensation, hope, etc. | 24,044 |
| A-2 | Love, respect, like, etc. | 7124 |
| B-1 | Surprise, amazement, etc. | 2350 |
| C-1 | Anger, hate, spite, etc. | 13,037 |
| D-1 | Sorrow, pity, guilt, etc. | 8201 |
| D-2 | Anxiety, fear, etc. | 7816 |
| E-1 | Neutral | 8717 |

There are many approaches to estimating emotion from sentences using linguistic features [34–38]. These approaches often use machine learning methods such as neural networks for training the emotion estimators [39,40]. Since more than one emotion label can be annotated to the same sentence, our approach also uses neural networks, as they are

easy to apply to a multi-label classification task. For training features, we used the sentence distributed representation vector.

**Table 3.** Example of sentences in the emotion corpus.

| Type | Example | Emotion Label |
|---|---|---|
| Spoken sentences | You cheated! | C-1 |
| | My head may hurt. | D-1 |
| | Now, I will drink v (≧∇≦) v | A-1 |
| Emotional words/phrases | hardship | D-1 |
| | bluff | C-1 |
| | cozy | A-1 |

**Table 4.** Statistics of the emotion corpus.

| # of Examples | 61,802 |
|---|---|
| # of Words | 845,940 |
| # of Vocabularies | 43,444 |

In the preprocess for modeling the emotion estimation, the utterance sentences were converted into fixed dimension dense vectors. Before processing, the sentences were morphologically analyzed and split into word units. The vectorization algorithm is based on neural networks. This method is called sentence embedding. We used sentence2vec, which is an implementation of sentence embedding similar to Doc2Vec or Paragraph2Vec. The framework for the training of the paragraph vector is shown in Figure 1.

After vectorizing the data with annotated emotion labels following the method described above, the emotion estimation model was trained with deep neural networks by using the sentence vector data as training data. The structure of the networks is of the fully-connected, four hidden layers, feed-forward type. Parameters such as the unit numbers and the activation function in each layer are shown in Table 5. This emotion estimation model is referred to as "Model-1" in the sections that follow. To train Model-1, we did not use sentences annotated with the "E-1" (Neutral) label. Because a neutral emotion is not distinctive, we believed that it would be difficult to extract significant features. Another type of deep neural network was built, which allowed us to add "Neutral" as emotion category "E-1". The parameters, unit numbers, and the activation function in each layer are shown in Table 6. Hereinafter, this second emotion estimation model is referred to as "Model-2". Each layer is defined as a fully-connected layer. According to the results from several tests, the two models with different architectures achieved the best scores. Therefore, we decided to use these architectures to estimate emotion from the sentences. The parameter settings for these models, such as the number of units, activation function, and dropout rate, were determined by selecting the best-performing model through training and testing on a randomly selected supervised corpus. Categorical cross-entropy was used as the loss function and Adam was used as the optimization algorithm. For the network configuration of the proposed model and the combinations of parameters, we attempted several patterns and selected the one that resulted in the best model accuracy.

**Table 5.** Network structure and hyper parameters (Model-1).

| Layer | Num. of Units | Activation Function | Dropout Rate |
|---|---|---|---|
| Input | 500 | tanh | 0.5 |
| Hidden-1 | 128 | tanh | 0.5 |
| Hidden-2 | 256 | tanh | 0.5 |
| Hidden-3 | 512 | tanh | 0.5 |
| Hidden-4 | 256 | tanh | 0.5 |
| Output | 6 | softmax | – |

**Table 6.** Network structure and hyper parameters (Model-2).

| Layer | Num. of Units | Activation Function | Dropout Rate |
|---|---|---|---|
| Input | 500 | tanh | 0.0 |
| Hidden-1 | 200 | tanh | 0.0 |
| Hidden-2 | 500 | tanh | 0.1 |
| Hidden-3 | 1000 | – | 0.1 |
| Hidden-4 | 100 | – | 0.1 |
| Output | 7 | softmax | – |

To avoid the effects of bias in the training data, we applied data augmentation to the sentence distributed representations when we trained Model-2. For each example in the training data, we created 50 augmented data elements by randomly adding noise to the values of the dimensions. We added these data evenly to the training data for each emotion label.

### 3.3. Extraction of Similarity Pattern Feature

Figure 3 depicts the process of calculating the emotion and sentence similarity vectors from the target utterance and the previous two dialogues. The emotion was estimated based on the sentence vector in the calculation of the emotion similarity vector. The estimation results were output as emotion vectors, and the emotion similarity vector was calculated based on the cosine similarity between each emotion vector. The sentence similarity vectors were calculated directly from the sentence vectors. Breakdowns in the responses of dialogue systems due to a lack of emotional understanding are generally caused by differences between the latest emotion presented in the system's utterance and the emotion in the user's latest utterance. Accordingly, the following features were extracted for each system response:

- Similarity $esim_{t-1,t}^{u,s}$ between the emotion ($E_{t-1}^u$) present in the latest utterance of the user and the emotion ($E_t^s$) generated in the current system response.
- Similarity $esim_{t-1,t}^{s,s}$ between the emotion ($E_{t-1}^s$) present in the latest response by the system and the emotion ($E_t^s$) generated in the current system response.
- Similarity $esim_{t-1,t-1}^{u,s}$ between the emotions $E_{t-1}^u$ and $E_{t-1}^s$.

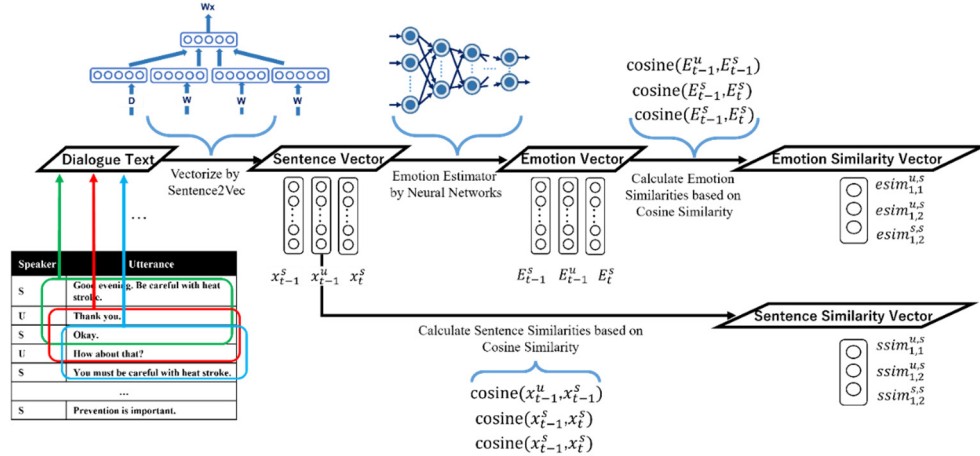

**Figure 3.** Extraction flow of emotion/sentence similarity vector.

Emotion ($E_{t-1}^u$, $E_t^s$, $E_{t-1}^s$) indicates the emotion level vector. We hold the values of the top two emotion labels and convert the values of the other emotion labels into zero.

Using a baseline method to compare against the proposed method, the following similarity vector features between the sentence distributed vectors were extracted:

- Similarity $ssim^{u,s}_{t-1,t}$ between the sentence distributed representation ($x^u_{t-1}$) that occurred in the latest user's utterance and the sentence distributed representation ($x^s_t$) of the current system response utterance.
- Similarity $ssim^{s,s}_{t-1,t}$ between the sentence distributed representation ($x^s_{t-1}$) that occurred in the previous system response utterance and the sentence distributed representation ($x^s_t$) of the current system response utterance.
- Similarity $ssim^{u,s}_{t-1,t-1}$ between the sentence distributed representations $S^u_{t-1}$ and $S^s_{t-1}$.

Rather than only using the system's current utterance or the user's prior utterance, using features calculated from the three utterances indicated in Figure 3 provided the system with more information on the context of the dialogue. The reason that even more utterances were not used is that we believed the system should detect a dialogue breakdown as early in the dialogue as possible and be able to quickly recover.

Table 7 shows an example of dialogue sentences and emotion similarity vectors based on the emotion estimation model using deep neural networks. In the figure, "$-1$" indicates that there were no annotations, as these lines were the utterance sentences of the user.

**Table 7.** Example of dialogue sentences and emotion similarity vectors.

| S/U | Utterance | Emotion Vector | Emotion Similarity (Top 2 Emotions) Vector | (O, X, T) |
|---|---|---|---|---|
| S | Good evening. Be careful with heat stroke. | A2:0.355 A1:0.330 C1:0.277 E1:0.038 | - | (30, 0, 0) |
| U | Thank you. | E1:0.965 D1:0.027 | - | (−1, −1, −1) |
| S | Okay. | A1:0.482 C1:0.264 A2:0.132 E1:0.121 | 0.00 0.00 0.60 | (11, 9, 10) |
| U | How about that? | E1:0.980 D1:0.020 | 0.00 0.00 1.00 | (−1, −1, −1) |
| S | You must be careful with heat stroke. | A1:0.728 A2:0.163 C1:0.088 E1:0.020 | 0.00 0.00 0.86 | (5, 11, 14) |
| U | Yes. And I got you a heat stroke at first sight. | E1:0.976 D1:0.024 | 0.00 0.00 1.00 | (−1, −1, −1) |
| S | Heat stroke is okay. | E1:0.900 A1:0.036 C1:0.027 A2:0.026 D1:0.011 | 0.00 1.00 0.04 | (8, 8, 14) |
| U | Which way are you talking about? | E1:0.951 D1:0.029 | 1.00 1.00 1.00 | (−1, −1, −1) |
| S | Prevention is important. | A1:0.889 A2:0.080 C1:0.031 | 1.00 0.00 0.04 | (6, 12, 12) |

### 3.4. Estimation of Dialogue Breakdown Label

We used canonical correlation analysis, a logistic regression, and recurrent neural networks to train the dialogue breakdown analyzer. Figure 4 shows the creation flow of dialogue breakdown regression model. The following sections describe each method.

Here, we propose a method for dialogue breakdown detection for the current utterance by training a model that simultaneously predicts dialogue breakdown labels for two system utterances using a multi-task learning neural network. It is known that multi-task learning neural networks can construct models efficiently and accurately by constructing networks that can learn multiple prediction targets at the same time and can share features such as relatedness.

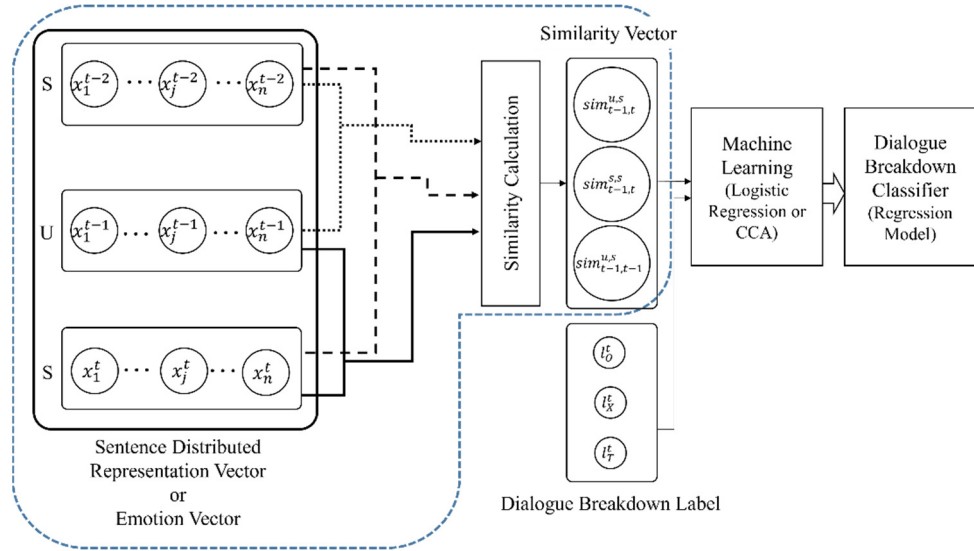

**Figure 4.** Creation flow of dialogue breakdown regression model.

In this model, we also propose an improved model in which the vector of emotion estimation results obtained from the emotion estimation model is added to the input.

Figure 5 shows the architecture of the multi-task neural network for dialogue breakdown detection. The input is a vector of distributed representations $X^t$ of two system utterances and one human utterance, and a vector of emotion labels $EV^t$ predicted from the system and three human utterances, respectively. The prediction targets are the dialogue breakdown labels $Z^{t-2}$ and $Z^t$ of the two input dialogue system utterances. The emotion vector used as input was that of Model-2.

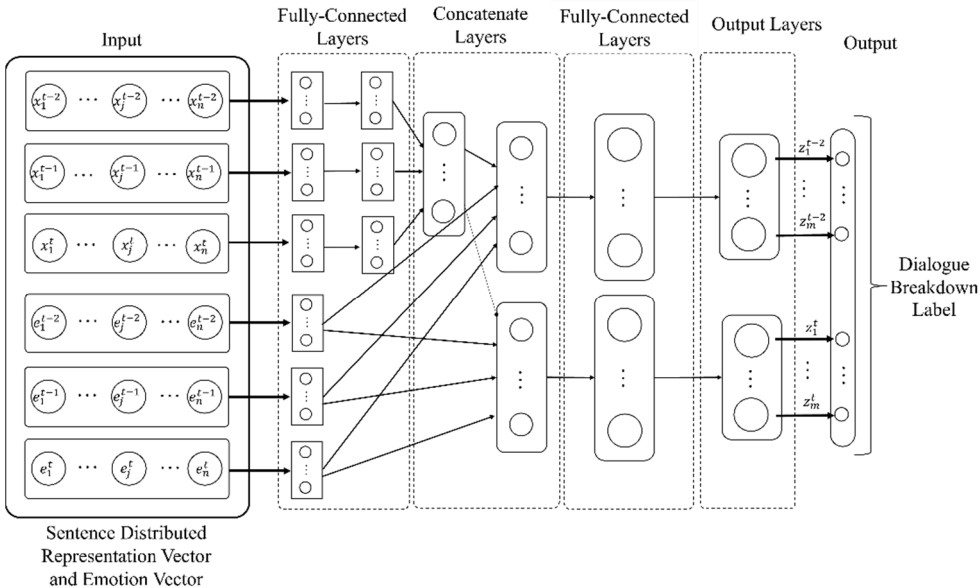

**Figure 5.** Multi-task learning of dialogue breakdown label.

The parameters of the neural network used in this architecture are shown below.

- Number of units of each layer

  Input layers: 500
  Middle Layers-1: 300, Middle Layers-2: 100
  Output Layers: $2 \times 2$ (binary) or $3 \times 2$ (three-valued)

- Dropout rate:

  Middle Layers-1: 0.4, Middle Layers-2: 0.1

- Activation function: softplus (Middle Layers-1, and 2), softmax (Output layers)
- Kernel initializer: Glorot's uniform
- Batch size: 1024
- Epochs: 200

Training and testing are performed using 10-fold cross-validation.

### 3.4.1. Canonical Correlation Analysis

The procedure for estimating the distributions of dialogue breakdown labels, $dv_i$, corresponding to the emotion similarity vectors, $ev_i$, can be applied to the problem of estimating other real-valued vectors, $dv_i$. In our study, canonical correlation analysis (CCA) [41–43] was used to estimate dialogue breakdowns. Canonical correlation analysis is a method to calculate correlation coefficients for multi-variates. It is also a method to calculate a coordinate system based on the five-fold cross covariance between two datasets.

The method is sometimes used to recommend information by calculating the correlation between different medium types, such as image and text, or music and biological information [44,45]. We used the CCA for estimating dialogue breakdown distribution from the similarity vector based on sentence distributed representation vectors or emotion level vectors.

### 3.4.2. Logistic Regression

In this study, we created a dialogue breakdown classifier by using logistic regression based on similarity patterns as features. Once the dialogue breakdown dataset was established, we used the dialogue breakdown labels with the maximum frequency as the answer labels, rather than using the distributions of the dialogue breakdown labels. We used logistic regression as a machine learning method, as proposed by Cox [46]. The model is a kind of statistic regression model. It is equivalent to simple perceptron; however, the methods to decide the parameters differ. We used the logistic regression model to train the similarity vectors based on either the sentence distributed representation vectors or the emotion level vectors in order to detect dialogue breakdown.

### 3.4.3. Recurrent Neural Networks

As our dialogue breakdown detection method using recurrent neural networks, we propose a method using a sequence of sentence vectors or emotion vectors as input features to estimate the dialogue breakdown labels. Three types of recurrent neural networks were used: Recurrent Neural Network (RNN) [47], Long Short-Term Memory (LSTM) [48,49], and Gated Recurrent Unit (GRU) [50]. The architectures of RNN, LSTM, and GRU are shown in Figures 6–8. Multi-layer deep neural networks such as RNN have the vanishing gradient problem. LSTM layers have a forget gate. The idea of the forget gate was proposed to solve the vanishing gradient problem. GRU layers have a long short-term memory network without an output gate. Therefore, the learning time is shorter than for the LSTM networks.

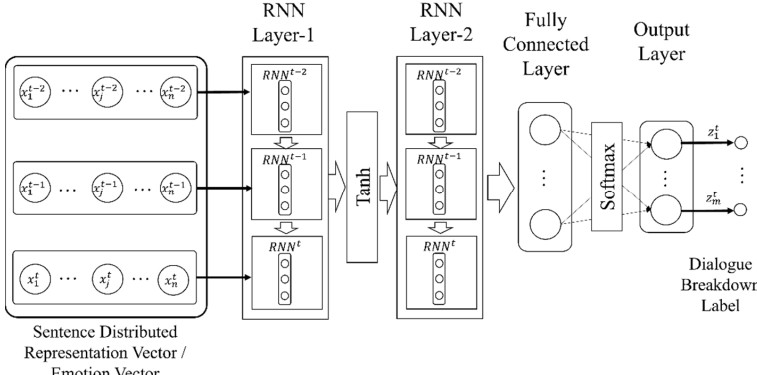

**Figure 6.** RNN network using sentence vector or emotion vector as feature.

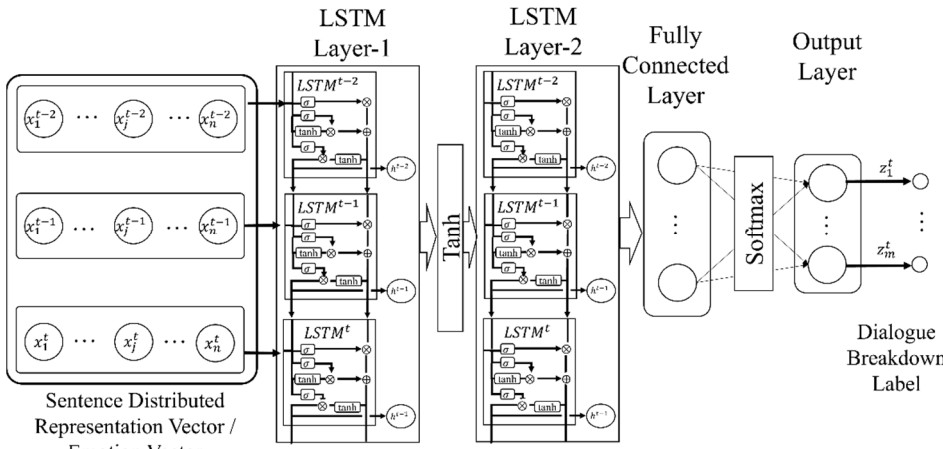

**Figure 7.** LSTM network using sentence vector or emotion vector as feature.

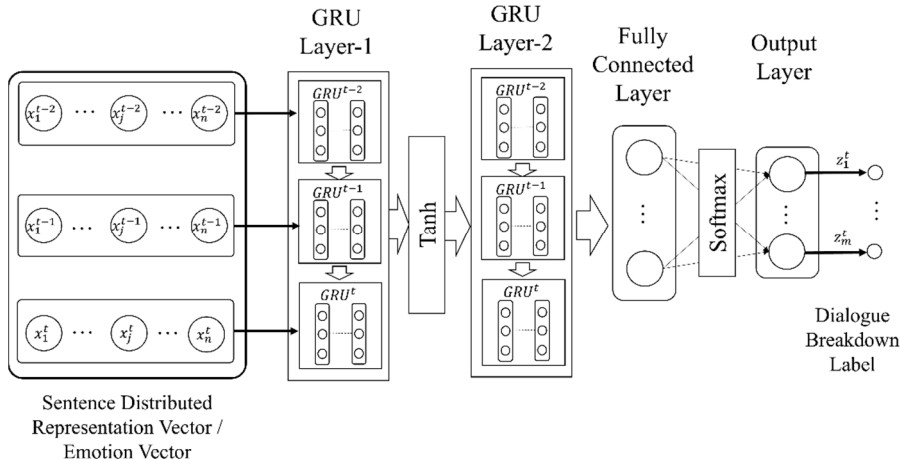

**Figure 8.** GRU network using sentence vector or emotion vector as feature.

## 4. Experiment and Results

In this section, we describe the experiments that were conducted to evaluate the validity of the proposed method. First, we describe the experimental data, experimental conditions, and evaluation method. Thereafter, we present the results of the evaluation experiments of the emotion estimation method and dialogue breakdown detection.

### 4.1. Experimental Data

We used the development (dev) dataset of Dialogue Breakdown Detection Challenge 2 as the experimental data. The dataset includes the system–user dialogue by following three types of dialogue system.

- DCM: chat dialogue system using NTT Docomo Chat Dialogue API [51];
- DIT: chat dialogue system by DENSO IT Laboratory inc. [52];
- IRS: example-based chat dialogue system based on IR-STATUS [53].

The numbers of dialogues and utterances for each system are shown in Table 8. We used the baseline method using sentence distributed vector as feature as a comparison target.

### 4.2. Evaluation of Emotion Estimation

To evaluate the performance of the two emotion estimation models, we compared the emotion estimation results from the two models with human judgment. Table 9 shows the number of participants in our test, along with their ages. Using more than 300 utterances (by the systems and the users), we asked each participant to describe the emotion involved by selecting from the list of emotions (joy, anger, sorrow, etc.) shown in Table 1. The

annotation agreement rate was calculated by using the Kappa coefficient value. Based on our calculations, the average Kappa value was 0.282. While this value is not especially high, we judged it not too low on the grounds that it is difficult to match seven kinds of annotation labels.

**Table 8.** Number of dialogues and utterances for each dialogue system.

| System | # of Dialogues | # of Utterances (System) | # of Utterances (User) | # of Words (System) | # of Words (User) |
|---|---|---|---|---|---|
| DCM | 50 | 550 | 500 | 4197 | 4179 |
| DIT | 50 | 550 | 500 | 15083 | 5204 |
| IRS | 50 | 550 | 500 | 8286 | 5341 |

**Table 9.** Number of subjects and their attributes.

| Gender | Age | Number |
|---|---|---|
| Male | 17 | 1 |
| | 19 | 1 |
| | 20 | 9 |
| | 38 | 1 |
| Female | 20 | 2 |

Normalized label frequency vectors annotated with the participants' responses were then used as emotion vectors. The precision, recall, and F-value for each label were calculated by comparing the results with the outputs from the two emotion estimation models. The top two emotion labels were then designated as the candidates for the output and the answer. Precision, recall, and F-value were used as the criteria in our evaluation. Table 10 presents the distribution of the emotion tags in the dialogue corpus used in the evaluation.

**Table 10.** Number of annotated tags for each dialogue system.

| Dialogue System | System/User | Emotion Tag (Code) | | | | | | |
|---|---|---|---|---|---|---|---|---|
| | | Joy (A-1) | Love (A-2) | Surprise (B-1) | Anger (C-1) | Sorrow (D-1) | Anxiety (D-2) | Neutral (E-1) |
| DCM | System | 270 | 188 | 77 | 37 | 31 | 74 | 320 |
| | User | 212 | 139 | 160 | 44 | 71 | 57 | 275 |
| DIT | System | 216 | 180 | 131 | 14 | 50 | 39 | 330 |
| | User | 177 | 149 | 251 | 18 | 91 | 40 | 236 |
| IRS | System | 214 | 201 | 159 | 23 | 92 | 50 | 247 |
| | User | 203 | 172 | 189 | 32 | 78 | 25 | 255 |

The experimental results are presented in Tables 11 and 12. As shown, the Model-2 results include the added E-1 ("neutral") emotion category; however, because Model-1 was trained without using the E-1 label, this category does not appear in the Model-1 results.

As can be seen, the average F-score for Model-1 was higher than that of Model-2; however, the B-1 values in Model-1 could not be estimated, since the results from the participant questionnaires included a large number of E-1 ("neutral") labels. Model-2 included "calm" as an output label type and could distinguish the types of emotions in more detail than Model-1. The fact that the dropout rate between the hidden layers was lower, and that the activation function was not used in hidden layers 3 and 4, are considered to be the reasons that Model-2 could produce relatively complete emotion estimation results. However, because the F-values were generally low and biased, it is difficult to determine which performance is better.

**Table 11.** Experimental results of emotion estimation (Model-1).

| Emotion | Precision | Recall | F-Value |
|---------|-----------|--------|---------|
| A-1 | 76.5 | 81.6 | 79.0 |
| A-2 | 59.3 | 85.1 | 69.9 |
| B-1 | 0.0 | 0.0 | 0.0 |
| C-1 | 19.1 | 60.8 | 29.1 |
| D-1 | 25.6 | 26.2 | 25.9 |
| D-2 | 17.3 | 22.6 | 19.6 |
| Average | 33.0 | 46.0 | 37.2 |

**Table 12.** Experimental results of emotion estimation (Model-2).

| Emotion | Precision | Recall | F-Value |
|---------|-----------|--------|---------|
| A-1 | 56.0 | 60.1 | 58.0 |
| A-2 | 39.8 | 50.3 | 44.4 |
| B-1 | 25.0 | 1.1 | 2.1 |
| C-1 | 8.6 | 63.0 | 15.2 |
| D-1 | 7.5 | 11.5 | 9.0 |
| D-2 | 6.7 | 9.5 | 7.8 |
| E-1 | 80.2 | 40.9 | 54.2 |
| Average | 32.0 | 33.8 | 27.2 |

Based on these results, Model-2 could estimate the E-1 ("neutral") labels with 80.2% precision. In fact, because many utterances did not express any particular emotion, Model-2, which could estimate "neutral", would appear to be generally more effective than Model-1 in estimating the emotion in utterances for the dialogue system.

*4.3. Evaluation of Dialogue Breakdown Detection*

To evaluate the effectiveness of the proposed method, we conducted an evaluation experiment. Dialogue data (50 pairs of dialogue data for development) were obtained from the three types of chat dialogue systems (DCM, DIT, IRS) and used as the dataset for Dialogue Breakdown Detection Challenge 2. We sorted the data according to each system and used the different datasets for the experiment.

The development data were only used to evaluate the data in a 50-fold cross-validation test by each dialogue unit. We compared the proposed method and the baseline method (s2v: using the similarity vector of the sentence distributed representation vector as the feature) using the following evaluation procedure:

- For the distribution of correct labels and the distribution of the labels that were outputted by the dialogue breakdown, compare

  — Cosine Similarity,
  — Jensen-Shannon Divergence (JSD) [54],
  — Mean Squared Error (MSE).

- Apply the detection rate of the dialogue breakdown response sentence (a rate that succeeded in estimating dialogue breakdown sentences correctly) (Accuracy).

Equation (1) shows the cosine similarity between the distribution of $P$ and $Q$. $P$ is defined as $(p_1, p_2, \ldots, p_i, \ldots, p_n)$, and $Q$ is defined as $(q_1, q_2, \ldots, q_i, \ldots, q_n)$.

$$\cos ine(P, Q) = \frac{\sum_{i=1}^{n} p_i \times q_i}{\sqrt{\sum_{i=1}^{n} p_i^2} \sqrt{\sum_{i=1}^{n} q_i^2}} \tag{1}$$

The Kullback–Leibler divergence [55] between distribution $f(x)$ and $g(x)$ is calculated by Equation (2). The Jensen–Shannon divergence for $D_{JS}(P||Q)$ and $D_{JS}(Q||P)$ is calculated by Equations (3) and (4) using $D_{KL}(f(x)||g(x))$.

$$D_{KL}(f(x)||g(x)) = \int f(x) \log\frac{f(x)}{g(x)} dx \tag{2}$$

$$D_{JS}(P||Q) = \frac{1}{2}D_{KL}\left(P||\frac{1}{2}(P+Q)\right) + \frac{1}{2}D_{KL}\left(Q||\frac{1}{2}(P+Q)\right) \tag{3}$$

$$D_{JS}(Q||P) = \frac{1}{2}D_{KL}\left(Q||\frac{1}{2}(Q+P)\right) + \frac{1}{2}D_{KL}\left(P||\frac{1}{2}(Q+P)\right) \tag{4}$$

Mean square error (MSE) is calculated by Equation (5).

$$\text{MSE} = \frac{1}{n}\sum_{i=1}^{n}(p_i - q_i)^2 \tag{5}$$

Larger cosine similarity values and higher detection rates, as well as lower JSD and MSE values, indicate better performance. JSD is used to measure the distance between some distributions. The MSE is used to measure the error between the vectors. These measures were also used in the Dialogue Breakdown Detection Challenge as measures of the distributional agreement. The cosine similarity, which is used to calculate the similarity between vectors, was used in this study, as well as the JSD and MSE, to examine the distributional agreement of the output labels.

However, because these indices do not measure the performance of dialogue breakdown detection, it was necessary to evaluate them using values including the accuracy, precision, recall, and F1-score.

The calculation formulas for the accuracy, precision, recall, and F-value (F1-score) are presented in Equations (6)–(9). $C_{match}$ indicates the number of labels output by the classifier that match the correct label. $C_{total}$ indicates the total number of correct labels in the evaluation dataset. The fit rate of label $l$ is denoted as $P_l$, the recurrence rate is denoted as $R_l$, and the F-value is denoted as $F_l$. $M_l$ represents the number of cases in which the correct answer is label $l$ out of the outputs with label $l$. $O_l$ indicates the number of cases predicted to have the label $l$. $C_l$ represents the number of cases in which the correct answer label is $l$.

$$Accuracy\ (\%) = \frac{C_{match}}{C_{total}} \times 100 \tag{6}$$

$$P_l(\%) = \frac{M_l}{O_l} \times 100 \tag{7}$$

$$R_l(\%) = \frac{M_l}{C_l} \times 100 \tag{8}$$

$$F_l = \frac{2 \times P_l \times R_l}{P_l + R_l} \tag{9}$$

As described earlier, 50 dialogues obtained from each of three dialogue systems, DCM, DIT, and IRS, were used as the experimental target.

Figure 9 shows the experimental results (Accuracy). In the figure, 'Model-1' and 'Model-2' indicate the results of the proposed methods, and 's2v' indicates the results of the baseline method. The symbol O means the label "Not a breakdown", the symbol T means "Possible breakdown", and the symbol X means "Breakdown".

For many combinations, better results were obtained by the method using emotion vectors than by the method using sentence vectors (s2v). When we used the similarity-based methods by Logistic Regression or CCA, we could obtain better results than the RNN-based methods (RNN, LSTM, GRU). We then evaluated binary classification using the following two patterns:

- [OT + X]: O and T are "Not breakdown", X is "Breakdown".
- [O + TX]: O is "Not breakdown", T and X are "Breakdown."

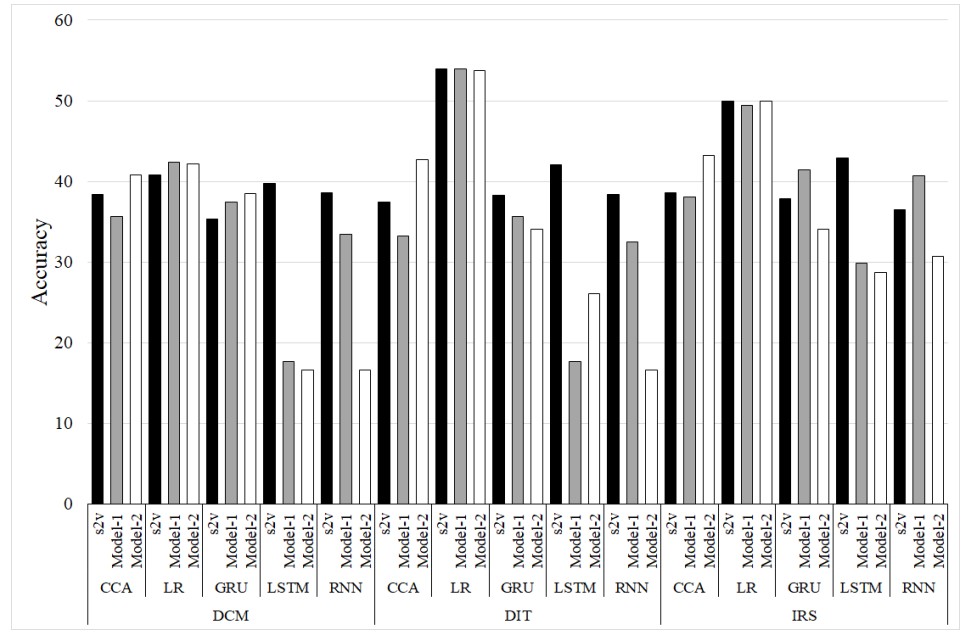

**Figure 9.** Experimental results (Accuracy).

These binary classification evaluations are used as the standard evaluation indicators. Recalls, precision, and F-values were calculated. Figure 10 shows the P–R curve for each result. As can be seen from the graphs, we found that our proposed method using emotion level vectors and neural networks produced better precision for the binary dialogue breakdown label classification task than the baseline method. However, the recalls for our proposed method were lower than those of the baseline method.

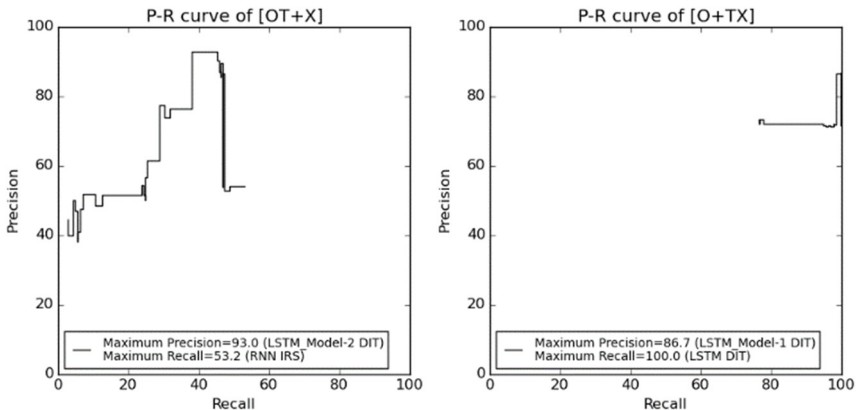

**Figure 10.** P–R curve of binary classification evaluation results (OT + X) and (O + TX).

The comparisons of breakdown detection accuracy and precision, recalls, and F-values for competition 2 are shown in Tables 13–15. In the table, P indicates precision, R indicates recall, and F indicates F-value. In addition, (X) indicates binary classification [OT + X] and (XT) indicates binary classification [O + TX]. The other detection methods for comparison are indicated in Table 16.

As shown, our proposed methods are able to detect [O + TX] labels in the system DCM and DIT with a higher F-value than the other methods. However, the accuracy of our proposed methods was generally lower than the other methods. Consequently, we believe that the emotion level vector is a good feature for detecting dialogue breakdown, even though it is just a simple six- or seven-dimension vector used as a training feature.

**Table 13.** Comparison with the other methods presented in DBDC2 (DCM).

| Method | Feature | Accuracy | P(X) | R(X) | F(X) | P(XT) | R(XT) | F(XT) |
|---|---|---|---|---|---|---|---|---|
| | s2v | 37.9 | 38.1 | 5.6 | 9.8 | 71.5 | 96.9 | 82.3 |
| RNN | Model-1 | 42.9 | 48.6 | 12.6 | 20.0 | 72.0 | 95.0 | 81.9 |
| | Model-2 | 36.5 | 40.0 | 4.2 | 7.6 | 71.3 | 96.4 | 81.9 |
| | s2v | 35.3 | 47.1 | 5.6 | 10.0 | 71.7 | 95.5 | 81.9 |
| LSTM | Model-1 | 39.8 | 40.9 | 6.3 | 10.9 | 71.4 | 97.8 | 82.5 |
| | Model-2 | 38.6 | 44.4 | 2.8 | 5.3 | 71.6 | **100.0** | **83.5** |
| | s2v | 38.3 | 47.6 | 7.0 | 12.2 | 71.2 | 96.1 | 81.8 |
| GRU | Model-1 | 42.1 | 51.7 | 10.5 | 17.4 | 72.2 | 97.8 | 83.0 |
| | Model-2 | 38.4 | 50.0 | 4.9 | 8.9 | 71.9 | 98.6 | 83.2 |
| Multi-task | s2v | 43.0 | 36.0 | 14.0 | 20.0 | 64.0 | **100.0** | 78.0 |
| Learning | s2v_Model-2 | 38.0 | 24.0 | 30.0 | 26.0 | 64.0 | 100.0 | 78.0 |
| HCU_run3 | | 50.4 | 52.0 | 29.2 | 37.4 | 91.0 | 39.6 | 55.1 |
| smap_run1 | | 41.5 | 43.4 | 47.8 | 45.5 | 73.9 | 90.8 | 81.5 |
| RSL16BD_run1 | | 40.5 | 0.0 | 0.0 | 0.0 | **100.0** | 0.0 | 0.0 |
| NTTCS_run1 | | 52.7 | 47.7 | 65.2 | 55.1 | 84.2 | 71.0 | 77.0 |
| NTTCS_run2 | | **56.5** | **52.3** | 58.4 | 55.2 | 87.5 | 62.4 | 72.8 |
| NTTCS_run3 | | 52 | 47.8 | 65.7 | **55.3** | 83.0 | 70.5 | 76.2 |
| KIT16_run2 | | 45.5 | 45.3 | 57.3 | 50.6 | 75.3 | 82.5 | 78.7 |
| OKSAT_run3 | | 35.5 | 38.2 | 85.4 | 52.8 | 73.6 | 81.6 | 77.4 |
| kanolab_run1 | | 43.3 | 37.2 | **92.7** | 53.1 | 72.7 | 90.0 | 80.4 |

**Table 14.** Comparison with the other methods presented in DBDC2 (DIT).

| Method | Feature | Accuracy | P(X) | R(X) | F(X) | P(XT) | R(XT) | F(XT) |
|---|---|---|---|---|---|---|---|---|
| | s2v | 41.5 | 85.6 | 46.3 | 60.1 | 86.6 | 99.8 | 92.7 |
| RNN | Model-1 | 29.8 | 76.5 | 38.0 | 50.8 | 86.7 | **100.0** | **92.9** |
| | Model-2 | 40.7 | 86.6 | 47.3 | 61.2 | 86.7 | **100.0** | **92.9** |
| | s2v | 37.4 | 89.7 | 46.8 | 61.5 | 86.6 | **100.0** | 92.8 |
| LSTM | Model-1 | 17.7 | 73.9 | 31.7 | 44.4 | 86.7 | **100.0** | **92.9** |
| | Model-2 | 33.4 | **93.0** | 45.4 | 61.0 | 86.7 | **100.0** | **92.9** |
| | s2v | 35.7 | 87.2 | 46.3 | 60.5 | 86.6 | **100.0** | 92.8 |
| GRU | Model-1 | 17.7 | 77.5 | 30.2 | 43.5 | 86.7 | **100.0** | **92.9** |
| | Model-2 | 32.5 | 90.4 | 45.9 | 60.8 | 86.7 | **100.0** | **92.9** |
| Multi-task | s2v | 48.0 | 11.0 | 2.0 | 3.0 | 77.0 | **100.0** | 87.0 |
| Learning | s2v_Model-2 | 41.0 | 23.0 | 27.0 | 25.0 | 77.0 | **100.0** | 87.0 |
| HCU_run1 | | 62.2 | 65.2 | 81.4 | 72.4 | 90.1 | 74.8 | 81.7 |
| HCU_run3 | | 62.4 | 65.5 | 81.8 | 72.7 | **90.4** | 84.2 | 87.2 |
| smap_run1 | | 58.4 | 59.7 | 81.8 | 69.0 | 84.7 | 99.3 | 91.4 |
| RSL16BD_run1 | | 59.1 | 54.0 | **100.0** | 70.1 | 84.3 | **100.0** | 91.5 |
| NTTCS_run1 | | 64 | 61.5 | 93.9 | 74.4 | 89.5 | 91.0 | 90.3 |
| NTTCS_run2 | | **65.5** | 63.2 | 94.3 | **75.7** | 90.0 | 89.1 | 89.5 |
| KIT16_run2 | | 59.1 | 59.4 | 83.7 | 69.5 | 86.5 | 93.2 | 89.7 |
| OKSAT_run1 | | 58.9 | 54.2 | 99.2 | 70.1 | 84.3 | 98.8 | 90.9 |
| kanolab_run1 | | 57.1 | 53.2 | 98.9 | 69.1 | 82.3 | 98.1 | 89.5 |

**Table 15.** Comparison with the other methods presented in DBDC2 (IRS).

| Method | Feature | Accuracy | P(X) | R(X) | F(X) | P(XT) | R(XT) | F(XT) |
|---|---|---|---|---|---|---|---|---|
| | s2v | 34.1 | 54.2 | 53.2 | 53.7 | 73.4 | 77.9 | 75.6 |
| RNN | Model-1 | 28.7 | **61.5** | 28.8 | 39.2 | 71.6 | **100.0** | **83.5** |
| | Model-2 | 30.7 | 50.0 | 24.9 | 33.2 | 71.6 | **100.0** | **83.5** |
| | s2v | 38.5 | 53.9 | 46.8 | 50.1 | 72.2 | 76.8 | 74.4 |
| LSTM | Model-1 | 16.6 | 54.3 | 24.4 | 33.7 | 71.6 | **100.0** | **83.5** |
| | Model-2 | 16.6 | 51.6 | 23.9 | 32.7 | 71.6 | **100.0** | **83.5** |

**Table 15.** *Cont.*

| Method | Feature | Accuracy | P(X) | R(X) | F(X) | P(XT) | R(XT) | F(XT) |
|--------|---------|----------|------|------|------|-------|-------|-------|
| | s2v | 34.1 | 52.9 | 48.8 | 50.8 | 73.5 | 76.5 | 75.0 |
| GRU | Model-1 | 26.1 | 56.5 | 25.4 | 35.0 | 71.6 | **100.0** | **83.5** |
| | Model-2 | 16.6 | 51.5 | 24.9 | 33.6 | 71.6 | **100.0** | **83.5** |
| Multi-task | s2v | 35.0 | 17.0 | 51.0 | 26.0 | 67.0 | 95.0 | 78.0 |
| Learning | s2v_Model-2 | 36.0 | 18.0 | 56.0 | 27.0 | 66.0 | 94.0 | 77.0 |
| HCU_run1 | | 53.1 | 56.7 | 58.9 | 57.7 | 77.7 | 53.8 | 63.6 |
| smap_run1 | | 42.0 | 48.3 | 54.1 | 51.0 | 72.5 | 98.3 | **83.5** |
| RSL16GBD_run2 | | 55.1 | 49.4 | **96.1** | 65.3 | 73.9 | 93.0 | 82.4 |
| RSL16GBD_run3 | | 55.3 | 49.7 | **96.1** | 65.5 | 74.0 | 92.7 | 82.3 |
| NTTCS_run2 | | **58.4** | 55.4 | 80.1 | 65.5 | **79.1** | 77.3 | 78.2 |
| NTTCS_run3 | | 58.4 | 53.9 | 84.0 | **65.7** | 78.9 | 82.6 | 80.7 |
| KIT16_run2 | | 49.8 | 50.6 | 68.8 | 58.3 | 75.0 | 85.7 | 80.0 |
| OKSAT_run1 | | 53.1 | 48.7 | 88.3 | 62.8 | 74.0 | 86.8 | 79.9 |
| OKSAT_run3 | | 45.1 | 50.1 | 86.1 | 63.4 | 75.6 | 84.0 | 79.6 |

**Table 16.** List of the other detection methods for the comparison.

| Name | Method | Difference for Each Run |
|------|--------|-------------------------|
| HCU_run1 | RNN, Multi-Layer Perceptron | Minimize MSE |
| HCU_run3 | | Minimize MSE and average of four models |
| smap_run1 | Neural Conversation Model, SVM | Use both outputs of Encoder/Decoder |
| RSL16BD_run1 | Word2Vec | Use breakdown rate of the develop data |
| RSL16BD_run2 | | Use breakdown rate for each pattern |
| RSL16BD_run3 | | Combination of run1 and run2 |
| NTTCS_run1 | Extra Trees Regression | Change features and training data |
| NTTCS_run2 | | |
| NTTCS_run3 | | |
| KIT16_run2 | Multi-Layer Perceptron, LSTM, Recurrent Convolutional Neural Networks | Type of dialogue breakdown was not used |
| OKSAT_run1 | Rule | Change rule |
| OKSAT_run3 | | |
| kanolab_run1 | Word2Vec, Rule | Liberalize threshold for judgment |

Next, we compared the three indices: cosine similarity (cos), Jensen–Shannon divergence (JSD), and mean square error (MSE). Tables 17–19 show the comparison of cosine, JSD, and MSE.

**Table 17.** Comparison of cosine, JSD, and MSE (DCM).

| ML | Feature | Cos | JSD | MSE |
|----|---------|-----|-----|-----|
| | s2v | 0.2306 | 1.5881 | 13.9775 |
| CCA | Model-1 | 0.5165 | 0.6127 | 1.6238 |
| | Model-2 | 0.5335 | 0.6431 | 0.5197 |
| | s2v | 0.8116 | 0.1727 | 0.1305 |
| LR | Model-1 | 0.8113 | 0.1735 | 0.1308 |
| | Model-2 | 0.8101 | 0.1735 | 0.1310 |
| | s2v | 0.8189 | 0.1650 | 0.1275 |
| RNN | Model-1 | 0.8276 | 0.1598 | 0.1232 |
| | Model-2 | 0.8221 | 0.1630 | 0.1261 |

**Table 17.** *Cont*.

| ML | Feature | Cos | JSD | MSE |
|---|---|---|---|---|
| | s2v | 0.8229 | 0.1628 | 0.1260 |
| LSTM | Model-1 | 0.8275 | 0.1604 | 0.1241 |
| | Model-2 | 0.8270 | 0.1610 | 0.1249 |
| | s2v | 0.8240 | 0.1623 | 0.1256 |
| GRU | Model-1 | 0.8297 | 0.1590 | 0.1230 |
| | Model-2 | 0.8259 | 0.1614 | 0.1250 |

**Table 18.** Comparison of cosine, JSD, and MSE (DIT).

| ML | Feature | Cos | JSD | MSE |
|---|---|---|---|---|
| | s2v | 0.4470 | 0.8176 | 2.4040 |
| CCA | Model-1 | 0.5774 | 0.4688 | 0.8583 |
| | Model-2 | 0.5242 | 0.7380 | 0.7088 |
| | s2v | 0.8592 | 0.1326 | 0.1051 |
| LR | Model-1 | 0.8603 | 0.1326 | 0.1048 |
| | Model-2 | 0.8578 | 0.1346 | 0.1056 |
| | s2v | 0.8781 | 0.1211 | 0.0985 |
| RNN | Model-1 | 0.8750 | 0.1247 | 0.1014 |
| | Model-2 | 0.8784 | 0.1219 | 0.0982 |
| | s2v | 0.8838 | 0.1181 | 0.0966 |
| LSTM | Model-1 | 0.8742 | 0.1254 | 0.1028 |
| | Model-2 | 0.8804 | 0.1212 | 0.0983 |
| | s2v | 0.8819 | 0.1192 | 0.0974 |
| GRU | Model-1 | 0.8740 | 0.1256 | 0.1029 |
| | Model-2 | 0.8798 | 0.1212 | 0.0982 |

**Table 19.** Comparison of cosine, JSD, and MSE (IRS).

| ML | Feature | Cos | JSD | MSE |
|---|---|---|---|---|
| | s2v | 0.2618 | 0.9637 | 2.7551 |
| CCA | Model-1 | 0.3812 | 0.8584 | 2.1685 |
| | Model-2 | 0.5060 | 0.6977 | 0.6770 |
| | s2v | 0.7828 | 0.1954 | 0.1375 |
| LR | Model-1 | 0.7821 | 0.1955 | 0.1377 |
| | Model-2 | 0.7826 | 0.1963 | 0.1376 |
| | s2v | 0.7028 | 0.2593 | 0.1773 |
| RNN | Model-1 | 0.8020 | 0.1802 | 0.1334 |
| | Model-2 | 0.7934 | 0.1848 | 0.1367 |
| | s2v | 0.7207 | 0.2447 | 0.1689 |
| LSTM | Model-1 | 0.8077 | 0.1775 | 0.1317 |
| | Model-2 | 0.8050 | 0.1788 | 0.1326 |
| | s2v | 0.7049 | 0.2556 | 0.1762 |
| GRU | Model-1 | 0.8069 | 0.1777 | 0.1318 |
| | Model-2 | 0.8012 | 0.1809 | 0.1341 |

Based on these results, we found the three methods (RNN, LSTM, GRU) effective in the distributional matching of the dialogue breakdown labels. When Model-1 was used as the emotion estimator, we obtained better results than for the method using other features, indicating that the emotion vector based on Model-1 is relatively effective in detecting dialogue breakdowns.

## 5. Discussions

As described in the previous section, our proposed approach using emotion level vectors as a feature produced good results in evaluations based on cosine, JSD, and MSE. However, several methods proposed in the latest dialogue breakdown detection tasks produced better results. Because our proposed method uses emotional information exclusively, it ignores important information such as topics, word similarities, and sentence similarities. It would seem, then, that a method that combines both emotion level vectors and sentence embedding should be developed.

Our proposed method achieved better results than the method using the simple vector feature of sentence embedding. Notably, the precision rate for our proposed method in the binary breakdown label classification task was better than the baseline method. Moreover, based on comparisons with other breakdown detection methods presented in DBDC2, our proposed method worked well in the binary breakdown label detection task [O + TX]. This suggests that the emotional feature is important in the dialogue breakdown detection task. We also believe that our method has the potential to extract important features such as the emotional differences between the system and the user (speaker), or the user's (speaker's) negative emotions caused by breakdowns.

The similarity features for the emotion level vectors were not very impressive, largely because the vector dimension was small (only three dimensions), and the similarities were simple values. However, if new features combining sentence similarities and emotion level similarities are applied to the dialogue breakdown detection task, we believe that the approach will produce significant improvements in performance.

Furthermore, the emotion estimation model that was used in this study was not very accurate when annotated limited dialogue data were used as the evaluation set. In the future, it is necessary to improve the accuracy by retraining and transfer learning the emotion estimation model using more accurate vectorization methods such as BERT, or by introducing models such as LSTM and CRF, which can consider the dialogue context and long-term emotions [56–58].

The proposed method first uses a model that is trained by a neural network on a supervised corpus with emotion labels of a certain size, and then uses the probability values of the emotion labels that are predicted by the model as emotion vectors. That is, it combines and executes the output results of multiple models in a manner similar to connecting them in a pipeline. Therefore, it is necessary to extract feature vectors from the middle layer to share the parameters of the learned emotion vectors. In the future, we would like to consider sharing parameters by learning multitasking between the estimation of the emotion vectors and the estimation of the dialogue breakdown for efficient performance improvement.

## 6. Conclusions

In this study, we hypothesized that "lack of emotion recognition", which is thought to be a factor in dialogue breakdown, would appear in differences of emotion expression tendencies. Therefore, we proposed a method to estimate the distribution of dialogue breakdown labels by extracting similarities between the emotion estimation results of utterances made immediately before the target utterance and using these as a feature.

In the evaluation experiment, we found that the proposed method proved largely superior to the method using the similarity of sentence distributed vectors as the feature. In future work, we intend to investigate whether "lack of emotional understanding" occurs in the dialogue breakdown response sentences. More specifically, we will develop a method to determine how the dialogue system behaves when predicting the other party's emotions before speaking, and quantitatively evaluate its performance using questionnaires. Furthermore, we would then use this method to analyze the consistency between dialogue breakdown and the presence of emotional understanding.

In analyzing performance, we found that in some instances, having only a few cases had a negative influence on results, as the method was forced to estimate an emotion



based on a small number of occurrences of the emotion. In this paper, we used an emotion estimation model that was not specifically designed for dialogues. In the future, we intend to construct a specialized emotion estimation model for a dialogue system where emotion labels are manually annotated. Moreover, we plan to improve detection accuracy by combining our method with the baseline method used in the dialogue breakdown detection challenge or with a topic-similarity-based dialogue. Furthermore, we would construct a dialogue breakdown detector with improved emotional recognition using models that consider context, such as BERT and Transformer, which have been frequently used in recent years.

**Author Contributions:** Conceptualization, K.M.; data curation, K.M.; funding acquisition, K.M. and M.S.; methodology, K.M.; supervision, K.K. and F.R.; validation, M.S. and M.Y.; visualization, K.M.; writing—original draft, K.M.; writing—review and editing, K.M., M.Y. and F.R. All authors have read and agreed to the published version of the manuscript.

**Funding:** This work was partially supported by the 2021 SCAT Research Grant and Grants-in-Aid for Scientific Research KAKENHI JP19K12174 and JP20K12027.

**Institutional Review Board Statement:** Not applicable.

**Acknowledgments:** We would like to thank the people who helped us annotate the emotion labels of the dialogues in this study.

**Conflicts of Interest:** The authors declare no conflict interest.

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
