# Peer review of "Emotion Analysis and Dialogue Breakdown Detection in Dialogue of Chat Systems Based on Deep Neural Networks"

_electronics, doi:10.3390/electronics11050695_

Round 1

Reviewer 1 Report

The research in this paper addresses the problem of dialogue rupture due to sentiment deficit by proposing a method for estimating the distribution of dialogue decomposition labels.The similarity between the sentiment estimation results prior to the extraction of the target discourse is used as a feature. In their experiments, the authors find that the proposed method largely outperforms the method that features the similarity of sentence distribution vectors. There are some comments that should be considered in the manuscript, as follows:
1. The structure of this manuscript is hard to read, and the English literature organization is a bit obscure and difficult to understand. Thus, the readability of the manuscript should be improved.
2. In Section 2.1. Insufficient elaboration of the principle of Method of Dialogue Breakdown Detection.
3. In Section 3.3. Extraction flow of emotion similarity vector‘s elaboration is not detailed enough not specific enough, you can say a little more clearly .
4. In Section 3.3. model-2 whether some of the set neural network parameters can be reflected in the network structure.
5. In Section 4.2. the accuracy of model-2 is obtained higher than model-1, and the article proves it with several tables, but the specific details of the experiment are less elaborated, please add them appropriately.
6. In Section 4.3. Several formulas are listed in the assessment of conversation break detection, and the article lacks sentences explaining the relevance of these formulas.
7. The size difference between the table and the image in the article is too large, so we suggest reducing the size of the table and coordinating it with the content of the text and the image.

Reviewer 2 Report

The present manuscript addresses an interesting question which is very relevant for the analysis of internet chats: where and how can we identify breakdowns which are emotionally motivated. The authors use a clear rationale, a good theoretical approach and employed a promising machine learning technique.

First of all, I am NOT a Machine Learning expert and so I will focus my review on communication and cognition aspects of the paper; furthermore I checked the paper for consistency, format and language expression.

For me there are some open questions:

  • How have the network parameters been set—on which theoretical basis or on which empirical evidences from previous research?
  • What does “simple perception” means in this context? (line 380)
  • I clearly missed a basis emotional model and specifically how emotions develop and how long an emotional state will be assumed to persist to be still relevant for being taken in account for similarity analysis with upcoming emotional inducement.
  • The qualification of single utterances regarding emotional status is possible, but it is much more difficult to take into account a longer termed picture, so isn’t it relevant if several consistent emotional expressions are stated?
  • The sample is VERY selectively taken, in fact only one young women compared with only 4 men of different ages—but mostly being very young. And we also observe a very low agreement rate, so is the whole project feasible at all as even very similar humans evaluate emotional situations VERY different.

Still, especially the finding with the lack of emotion recognition as a driver for dialogue breakdown is important to be followed further.

Round 2

Reviewer 1 Report

The research in this paper addresses the problem of dialogue rupture due to sentiment deficit by proposing a method for estimating the distribution of dialogue decomposition labels. The similarity between the sentiment estimation results prior to the extraction of the target discourse is used as a feature. In their experiments, the authors find that the proposed method largely outperforms the method that features the similarity of sentence distribution vectors. There are some comments that should be considered in the manuscript, as follows:

  1. The picture and the text of the article are not in harmony, please adjust.
  2. In Section 6, you need to adjust the expressiveness of the text.
